# Bioactive Molecules from *Myrianthus arboreus*, *Acer rubrum*, and *Picea mariana* Forest Resources

**DOI:** 10.3390/molecules28052045

**Published:** 2023-02-22

**Authors:** Martha-Estrella García-Pérez, Pierre-Betu Kasangana, Tatjana Stevanovic

**Affiliations:** 1Faculty of Chemistry-Pharmacobiology, Michoacana University, Morelia 58240, Michoacán, Mexico; 2SEREX, College Centre for Technology Transfer Affiliated with Rimouski Cégep, Québec, QC G5J1K3, Canada; 3Renewable Materials Research Center (CRMR), Department of Wood Sciences and Forestry, Université Laval, Québec, QC G1V0A6, Canada

**Keywords:** antioxidant, extractive, forest, nutraceutical, polyphenol, terpenes

## Abstract

Forest trees are the world’s most important renewable natural resources in terms of their dominance among other biomasses and the diversity of molecules that they produce. Forest tree extractives include terpenes and polyphenols, widely recognized for their biological activity. These molecules are found in forest by-products, such as bark, buds, leaves, and knots, commonly ignored in forestry decisions. The present literature review focuses on in vitro experimental bioactivity from the phytochemicals of *Myrianthus arboreus*, *Acer rubrum*, and *Picea mariana* forest resources and by-products with potential for further nutraceutical, cosmeceutical, and pharmaceutical development. Although these forest extracts function as antioxidants in vitro and may act on signaling pathways involved in diabetes, psoriasis, inflammation, and skin aging, much still remains to be investigated before using them as therapeutic candidates, cosmetics, or functional foods. Traditional forest management systems focused on wood must evolve towards a holistic approach, allowing the use of these extractives for developing new value-added products.

## 1. Introduction

Forests are considered one of the most important natural resources globally, not only for their undeniable ecological importance but also for their wealth of bioactive molecules. Wood-based materials include solid wood products, engineering composites, paper, and fiber products. Forest exploitation generates large amounts of by-products (leaves, twigs, knots, buds, bark, acorns, etc.) considered an important resource for obtaining antioxidant and bioactive molecules [1].

The demand for such molecules of natural origin for the food, cosmetic, and pharmaceutical industries has increased in recent years due to the perception of their lower toxicity and reduced healthcare costs regarding synthetic products [2,3]. Among the woody species, the differences in the chemical structures of the three main components that form the cell walls (cellulose, hemicelluloses, and lignins) are relatively small, especially when compared with the great diversity of the components extraneous to the cell walls, commonly designated as extractives, as they can be conveniently solubilized by organic solvents and/or water. Even though the extractives are usually present in low proportions compared with the cell wall components, many wood properties, such as color, odor, natural durability, density, etc., are related to the type and quantity of extractives [4].

There are many types of chemical substances belonging to the forest tree extractives, including terpenoids and polyphenols, widely recognized for their antioxidant activity. Heartwood formation, one of the distinctive metabolic events of woody plants, involves the deposition of biologically important polyphenols, such as stilbenes, flavonoids, proanthocyanidins, lignans, and neolignans. These extractable “woody polyphenols” have a structural analogy with lignins, due to their common phenylpropanoid biosynthetic pathway [1]. As ubiquitous plant constituents, polyphenols are important for the human diet, being present in vegetables and fruits, as well as in medicinal herbaceous plants used in folk medicine. The potential application of these molecules as natural antioxidants is related to their chemical structure, which interferes with different phases of oxidative reactions in the organism. The number and position of the hydroxyl groups and the presence of aromatic rings allow them to react with oxygen species, also chelating ferric and other cations by transferring hydrogen atoms or by donating an electron to the radicals. However, their action goes beyond their antioxidant activity since these molecules and their metabolites can act on signaling pathways involved in the pathogenesis of diseases, such as cancer, inflammation, brain neuromodulation, Alzheimer’s disease, diabetes, and psoriasis [5,6].

One of the most remarkable cases illustrating the presence of bioactive phenols in tree residues is that of Salix spp. The story of Salix bark constituents as analgesic agents goes deeply into the times of ancient Egyptians, as it was revealed by a discovery, in the mid-19th century, of two ancient papyrus scrolls dated 1500. B.C, one of which described the use of willow barks [7]. These discoveries inspired the identification of salicin, an important glucoside of salicylic acid, as the component of Salix bark by the very end of the 19th century. Further salicin acetylation allowed the synthesis of aspirin, one of the most widely used medicines [7].

Recognized representatives of bioactive polyphenols in foods include resveratrol in red wine, epigallocatechin gallate in green tea, chlorogenic acid in coffee, anthocyanins in colored fruits, and procyanidins in grape seed, which have been important research objects for food science and nutrition [8]. In forest trees, the preponderance of such phenolic compounds is particularly remarkable in bark and knot wood [9,10]. For example, *Picea abies* knots contain approximately 6–24% of lignans, with the 7-hydroxymatairesinol as the dominant one [11], whereas *Picea mariana* bark is a natural source of resveratrol [12]. In the context of the wood-processing industry, this is of great importance, since these are abundant by-products from wood transformation with scarce utilization.

In addition to polyphenols, mono-, di-, sesqui-, and triterpenoids have been long known and utilized as forest-tree-derived resins and balsams. Terpenoids protect the tree from the invasion of pathogens and herbivores by inducing multiple defense mechanisms. Although the general composition of terpenes is characteristic of each species, it can differ between two individual trees. Drought, temperature fluctuations, air, and soil pollution, or the attack of pathogens can cause a reorganization of biosynthesis and emission of terpenes from trees [4]. Even though conifers from Pinus, Picea, or Abies genera can store terpenes, they can also emit terpenes after their synthesis. Although volatile terpenes are typically emitted by softwoods, hardwoods also synthesize different terpenes, among which exist high amounts of nonvolatile terpenes [13]. This is the case of species from the *Eucalyptus*, *Cedrus*, *Quercus*, and *Betula* genera, recognized for their terpenic composition.

The antioxidant or prooxidant activity of a particular terpene depends on its doses and structural characteristics—at high concentrations, they can act as prooxidants, whereas at low concentration, they can function as antioxidants [14]. For the monoterpenes group, the hydrocarbon-type, containing a methylene group in its structure, and the oxygenated type, showing phenolic structures, have the highest antioxidant activity, while in the case of sesquiterpenes, allylic alcohol types are the most active as antioxidants [15]. The phenolic O-H included in some types of diterpenes, such as phenolic derivatives of abietane-type resin acids, seem to be important in the antioxidant activity of these compounds, mainly in their sequestering properties. Dienone–phenol rearranged triterpenes have the highest antioxidant activity among different structural types of triterpenes. Among tetraterpenes, carotenoids act as very efficient antioxidants, which is attributable to the conjugated double bonds of their chemical structure [15]. However, in most cases, there is no established relationship between the structure and antioxidant activity of these compounds. These molecules also have pleiotropic effects in cancer, diabetes inflammation, neuromodulation, and cardiovascular diseases, as they interfere with pathways involved in these diseases [16].

In forest trees, terpenoids are stored mainly in the resin of the heartwood and sapwood of conifers, as well as in the sapwood of hardwoods [17]. They are also found in the needles of conifers and leaves of hardwoods. In conifers, the terpenoid group consists mostly of monoterpenes (e.g., α-,β-pinene), sesquiterpenes (e.g., β-caryophyllene), and diterpene resin acids (e.g., abietic acid derivatives), whereas triterpenes (e.g., betulin and lupeol) and sterols (e.g., β-sitosterol) dominate in hardwoods [18,19]. One of the most well-known and commercially successful derivative of diterpenoids isolated from forest resources is paclitaxel, which has been identified, for the first time, from the bark of *Taxus brevifolia* [20]. Paclitaxel is also found in *Taxus canadensis* [21] and is used as a chemotherapeutic agent for treating various metastatic cancers, such as ovarian and breast cancers. Other triterpenoids found in forest trees are betulin and betulinic acid present in the outer bark of birch. In contrast to betulinic acid, betulin is present in significantly larger quantities in the birch outer bark (10–35%) [22]. Currently, betulin is used as a cosmetic ingredient but mostly as a precursor for the synthesis of betulinic acid, which is well-known for its antitumor, anti-inflammatory, and anti-HIV activities [23].

The modern agroforestry industry is based on a high volume–low value economical model which requires the processing of large quantities of raw material to be cost-effective [24]. The presence of bioactive molecules, particularly terpenes, and polyphenols in by-products (stems, bark, twigs, knots, etc.) available through eco-friendly extraction, could be a path for a transition from a traditional linear forest economy to a circular economy where undervalued forest coproducts and wood waste could be converted into valuable market products [25], mainly for food, cosmetic, or pharmaceutical industries. This approach provides opportunities for the forestry sector, allowing access to new markets and promoting the protection of the environment since by-products are a potential source of forest fires and pests [25].

*Myrianthus arboreus*, *Acer rubrum*, and *Picea mariana* are forest species important for the African and Canadian economies, respectively [26,27]. These trees are used as wood and firewood, also being employed as food (*M. arboreus*), beverage (*Picea mariana*), chewing gum (*Picea mariana*), and maple syrup (*A. rubrum*) and being used in traditional folk medicine [26,28,29]. In the last two decades, evidence has been gathered regarding the presence of antioxidant and bioactive molecules in extracts from forest by-products of these species [19,30,31,32,33]. However, few reviews are available attempting to link the presence of these compounds in *M. arboreus*, *A. rubrum*, and *P. mariana* by-products and their potential for developing new value-added products in the context of current forest practices.

This review provides an updated overview of the presence of bioactive molecules, particularly terpenes and phenols, identified in *Myrianthus arboreus*, *Acer rubrum*, and *Picea mariana* forest resources and by-products. These species were chosen considering their richness in bioactive compounds, their importance as forest resources in Africa and North America, and their utilization as food, beverage, and in traditional medicine [34,35,36,37]. Finally, the challenges and opportunities for a complete valorization of forest extracts for potential future applications, mainly in the food and cosmetic industries, are discussed considering existing procedures for wood transformation. 

## 2. Bioactive Molecules from Promising Forest Resources of *Myrianthus arboreus*, *Acer rubrum*, and *Picea mariana*

### 2.1. Myrianthus arboreus

*Myrianthus arboreus* P. Beauv. (Cecropiaceae) is a tree that grows in tropical areas of the West African rainforest. It is an edible indigenous woody plant important for the rural economy of the region since, in addition to its direct dietary benefits, it is used as timber, firewood, and traditional medicine, also having sociocultural and religious roles [26]. Several bioactive compounds, notably bioactive phenols and terpenes have been identified in extracts from forest by-products of this tree (Table 1).

*M. arboreous* root bark and stem bark are used by local communities to treat diabetes and its complications [19]. Diabetes mellitus is a metabolic syndrome characterized by hyperglycemia accompanied by alterations of fat, protein, and carbohydrate metabolism that results from defects in insulin secretion, reduced insulin action, or both [15]. During diabetes, high amounts of free radicals are generated from glucose oxidation and nonenzymatic glycation of proteins as well as an impairment of antioxidant enzymes, contributing to oxidative stress and the development of diabetic co-morbidities, such as cataracts, nephropathy, encephalopathy, and cardiovascular diseases [15]. In recent years, excellent reviews have been published connecting oxidative stress and diabetes [44,45,46], and some antioxidant strategies have been proposed to counteract it, including natural products [45]. The advantage of using natural products as therapeutic candidates in this disease is related to their low costs and multiplicity of action, which goes beyond their antioxidant capacity.

The ethyl acetate fraction (EAc) from the ethanolic extract (EtOH) of the stem bark of *M. arboreus* acts as an antioxidant in vitro, also stimulating glucose uptake in C2C12 myotubes and 3T3-L1 adipocytes by inhibiting alpha-glucosidase and alpha-amylase activity [47]. This extract also provoked a significant decrease in body weight, total protein, HDL-cholesterol, plasma glucose, LDL-cholesterol, triglycerides, serum urea, and serum creatinine in streptozotocin-induced diabetic rats [48]. Moreover, the oral administration of a methanolic extract of leaves from *M. arboreus* significantly reduced body weight gain, inflammation, basal glycemia, and insulin resistance in a mouse model of metabolic syndrome induced by a high-fat diet intake [43].

The antioxidant activity of the EtOH extract of *M. arboreous* root bark and its fractions were studied using DPPH free radical and ORAC assays [49]. No significant differences between the EAc fraction and Oligopin^®^, a commercial polyphenol-rich extract from maritime pine bark, were found. The EtOH extract, by contrast, exhibited the highest antioxidant capacity as determined by the phosphomolybdenium method [49]. The antidiabetic potential in vitro of this extract and its fractions to modulate glucose uptake in muscle C2C12 cells and to regulate glucose homeostasis was further investigated in cultured hepatocytes (H4IIE and HepG2 cell lines) [50]. In liver cells, EtOH root bark extract and its fractions reduced hepatocyte glucose-6-phosphatase (G6Pase) activity through mechanisms involving both the insulin-dependent Akt pathway and the insulin-independent pathway, implicating the phosphorylation of AMP-activated protein kinase (AMPK) [50]. Interestingly, ethyl acetate (EAc) fraction (IC_50_ = 14.1 µg/mL) induced a significant increase in the AMPK phosphorylation, while EtOH extract (IC_50_ = 4.83 µg/mL) and its hexane (IC_50_ = 5.89 µg/mL) fraction stimulated both the Akt and AMPK pathways. The G6Pase inhibitory activity of the plant was concentrated in EAc fraction (56.8% reduction, IC_50_ = 14.1 μg/mL), being close to that of insulin, a drug used for diabetes (61% reduction, 100 nM). Additionally, the EtOH extract from *M. arboreus* root bark (EC_50_ = 13.3 µg/mL) and Hex fraction (EC_50_ = 19.5 µg/mL) elicited a significant positive action on glycogen synthase (GS) activity in HepG2 cells hepatocytes, while EAc fraction (EC_50_ = 20.4 µg/mL) showed a moderate action. A high correlation was found between the ability of *M. arboreus* extract and fractions to phosphorylate the glycogen synthase kinase-3 (GSK-3) and their action on GS (R^2^ = 0.81, *p* < 0.01) [50].

The sub-fractionation of the EAc fraction led to the isolation of five phenolic compounds, including two regioisomers C-glycosyl flavone (isoorientin and orientin), along with chlorogenic acid, protocatechuic acid, and protocatechuic aldehyde. Other Δ^12^ ursene-type pentacyclic triterpenes containing the *trans*-feruloyl moiety (H1–4), along with five well-known terpenoids, were identified from the hexane fraction named 3β-*O*-*trans*-feruloyl-2α,19α-dihydroxyurs-12-en-28-oic acid (H1), 3β-*O*-*trans*-feruloyl-2α-hydroxy-19α-methoxyurs-12-en-28-oic acid (H2), 2α-acetoxy-3β-*O*-*trans*-feruloyl-19α-hydroxyurs-12-en-28-oic acid (H3), and 2α-acetoxy-3β-*O*-*trans*-(3′-methoxy-4′-formyl)cinnamoyl-19α-methoxyurs-12-en-28-oic acid (H4) [39]. Isoorientin, H3, and H2 were determined to have better antidiabetic potential in vitro (Figure 1a) [40]. More interestingly, H3, the main compound from EtOH extract, showed the most potent G6Pase inhibition and GS stimulation [39]. Altogether, this research demonstrated that isoorientin, H3, and H2 were responsible, at least in part, for the antidiabetic in vitro potential activity of *M. arboreus* root bark extract.

Although the in vitro antidiabetic potential of extracts from this tree has been studied more extensively in recent years, other investigations in animals and cell lines suggest that extracts derived from *M. arboreus* forest by-products, such as leaves and bark, also exhibit wound-healing [52], antimicrobial [53], and antinociceptive [54] properties. Moreover, aqueous and methanolic extracts from *M. arboreus* leaves induced an early puberty onset and an increased fertility rate in female Wistar rats [55].

As young leaves of this forest species are popularly consumed in West Africa as vegetable soup, their nutritive value was determined. The crude fiber content of the leaves (11.6% DW) was within the range of other edible vegetables, whereby they were considered a good source of crude protein (18.74% DW). However, the leaves also contain anti-nutritional factors, such as alkaloids, tannins, saponins, trypsin inhibitors, oxalic acids, and phytic acids [56]. Consequently, the preparation of functional foods derived from the leaves of *M. arboreus* must ensure that the proportions of these compounds are within the established limits in order to avoid complications with the assimilation of other nutrients, such as proteins, calcium, iron, etc.

### 2.2. Acer rubrum

Red maple (*Acer rubrum*) is a native maple of Northeastern American forests. Both the red and sugar maple (*Acer saccharum*) species are widely regarded for the quality of their wood and their sap, which can be used to produce maple syrup. Other tissues of the red maple species, such as buds, flowers, leaves, twigs, and stem/branch barks, are employed as folk medicine by the First Nation of Canada and Native Americans to treat/prevent various diseases, including skin ailments, diabetes, and inflammation [30].

Red maple bark extract (RMBE) functions as an antioxidant due to its ability to scavenge DPPH (EC_50_ DPPH = 5.02 µg/mL). Additionally, it has a high content of phenols (455 mg GAE/g) [57]. Indeed, when it was compared with an aqueous ethanol extract from *propolis*, a bee product used in the food industry, antioxidant values appeared of a similar order of magnitude. This comparable tendency held for cranberry pomace extract and cranberry fruit extracts, respectively, at 343.2 mg/g DPPH·(EC_50DPPH_) and 3690–10,050 µmol TE/g (ORAC). Finally, red wine extract exhibited an EC_50_ DPPH·value (1380 mg/g DPPH) close to the value for the optimized RMBE [57].

Anhydro-glucitol-core gallotannins (ACGs) represent a specific class of hydrolyzable tannins characteristic of red maple, particularly ginnalin A (P10), ginnalin 3,6 (P11), and ginnalin C (P9) (Figure 2b). Among twelve hydrolyzable gallotannins, some ACGs have been identified in RMBE [58]. Using the DPPH spiking test, three of the identified compounds (ginnalin A, ginnalin C, and gallic acid) were tested to demonstrate their antioxidant activity. The latter revealed that at 0.5 mg/mL, ginnalin A, a major phenolic compound of RMBE, exhibited a higher peak area (PA) reduction (87.9%) than those of ginnalin C (75.8%) and gallic acid (75.9%). These results confirmed the ability of these compounds to scavenge oxidant radicals, such as DPPH [58].

ACGs were evaluated for their effect on neutrophil viability using flow cytometry evolution. Neutrophils were chosen, as they are key mediators in chronic and acute inflammation [59]. At 100 μM, the three compounds significantly increased the rate of the late apoptotic cells. Ginnalin A and Ginnalin 3,6 had additive effects on neutrophil apoptosis while Ginnalin C is antagonistic to apoptosis induced by ginnalin A and Ginnalin 3,6. Overall, these results suggested that these isolated phenolic compounds can affect neutrophil functions, such as their programmed cell death [51,60]. More interestingly, to modulate the viability of human blood neutrophils, the three compounds specifically targeted the apoptotic proteins, such as FADD, phospho-Rad17, SMAC/Diablo, and cytochrome c [51]. A simplified view of major variations in the expression of apoptosis-related proteins of neutrophils treated by compounds is proposed in Figure 1b. Hence, in addition to their antioxidant potential, these ACGs could facilitate the resolution of neutrophil-mediated inflammatory diseases.

Extracts from leaves, stems/twigs, bark, and sapwoods of *A. rubrum* were evaluated for their antiproliferative effects against human colon tumorigenic and non-tumorigenic cells [61] All extracts showed a greater ability to inhibit the growth of the colon cancer cells compared with normal cells in vitro, and this ability increased with Ginnalin A levels. The *A. rubrum* leaf extract was considered to be the most active antiproliferative extract [61].

Another study showed that a standardized extract from *A. rubrum* leaves (Maplifa^TM^) exhibits anti-tyrosinase and anti-melanogenic effects [62]. The anti-melanogenic effect was attributed mainly to the presence of Ginnalin A and its ability to act in an additive, synergistic, and/or complementary manner with other minor compounds, such as Ginnalin B and Ginnalin C [62]. These results were in line with a previous investigation that demonstrated the anti-tyrosinase activity in vitro of the bark extract of this tree and its potential for cosmeceutical applications [63].

Anhydro-glucitol-core containing gallotannin-enriched red maple leaf extract (MLE) alleviated obesity in a mice model fed with a high-fat diet [64]. MLE contributed to the decrease in body weight, improving insulin resistance and reducing fat mass and ectopic accumulation of hepatic lipids. In addition, MLE improved inflammation of the liver and white adipose tissue of the animals. These beneficial effects were explained, at least in part, due to the modulation of the relative abundance of butyrate- and acetate-producing bacteria in the gut microbiota [64].

Overall, the results described above demonstrate the potential of the extracts of the bark, buds, and leaves of *Acer rubrum* as antioxidant, antiaging, antiproliferative, anti-inflammatory, and anti-obesogenic agents. However, subsequent clinical studies are still needed before proposing their use in humans.

### 2.3. Picea mariana

Black spruce (*Picea mariana* [Mill.] B.S.P.) is considered the most important commercial and reforestation species in eastern Canada. Its wood is highly appreciated for furniture, pulpwood, and lumber production, thereby generating by-products such as bark, leaves, and twigs with poor utilization. In recent years, some investigations have focused on the valorization of black spruce (BS) bark following multiple strategies, including chemical characterization and the study of the extract’s bioactivity to develop new natural products useful to counteract oxidative stress, aging, and psoriasis (Figure 3).

In a first attempt to establish the possible therapeutic utilization of polyphenolic bark extracts from this species for psoriasis, the antioxidant activity of their aqueous and ethanolic extracts was compared with those of yellow birch (*Betula alleghaniensis*) (YB), balsam fir (*Abies balsamea* (L.) Mill., BF), and jack pine (*Pinus banksiana* Lamb., JP). Additionally, their toxicological and antiproliferative activities on normal and psoriatic keratinocytes were determined [65]. The extract from *P. mariana* showed the highest scavenging capacity among the extracts obtained by hot water extraction, particularly towards hydrogen peroxide (EC_50_ = 48.30 ± 0.80 µg/mL) and superoxide radical (EC_50_ = 107.4 ± 1.73 µg/mL). Moreover, BS aqueous extract at 110 µg/mL inhibited, by 18 and 21%, the growth of normal and non-lesional keratinocytes, respectively. Consequently, the purification of such crude extract was performed to enhance its biological activity [65].

A further in vitro study demonstrated that the ethyl acetate soluble fraction (BS-EAcf) from the aforementioned crude aqueous extract inhibited the production of cytokines, chemokines, adhesion molecule ICAM-1, nitric oxide, prostaglandin E2, elafin, and VEGF produced by psoriatic keratinocytes under TNF-α activation through down-regulating the NF-κB pathway and without causing keratinocyte toxicity (Figure 4) [66]. BS-EAcf can inhibit immune pathways associated with TNF-α-induced acute and chronic inflammatory responses by psoriatic keratinocytes, suggesting that the molecules contained in the aqueous extract from *Picea mariana* bark had a therapeutic potential to treat psoriasis.

The TNF-α is an inflammatory molecule crucial for psoriasis pathophysiology, as shown by the significant antipsoriatic effect of biological molecules (infliximab, etanercept, and adalimumab), which block its action. The inflammation induced by TNF occurs through activation of the NF-κB pathway, after the binding of this cytokine to their receptors. The Picea mariana extract inhibits the IκBα degradation and migration of p50/p65 subunit to the psoriatic keratinocyte nucleus, thereby avoiding the transcription for many proteins involved in the psoriasis pathogenesis, such as IL-6, IL-8, CX3CL1/fractalkine, iNOS, elafin, VEGF, and ICAM-1.

The BS-EAcf was chemically characterized by HPLC, NMR, and MS analyses. Major compounds are shown in Figure 2a. It was estimated that the initial BS dry bark contained at least 104 µg/g of *trans*-resveratrol (P2), a higher content than that of other edible sources, such as dark chocolate (0.4 µg/g) and peanuts (0.03–0.14 µg/g), so BS bark was considered to be a profitable source of this molecule [12]. Resveratrol is a stilbene widely recognized as antioxidant, able to protect against obesity, diabetes, cardiovascular pathologies, cancer, and osteoporosis [67].

In an additional analysis using a combination of ultraviolet (UV) profiles, mass spectra, and RMN, the following compounds were isolated from the hot water bark extract of *P. mariana*: *trans*-p-coumaric acid β-D-glucopyranoside and *trans*-ferulic acid β-D-glucopyranoside [68]. The hot water extract also contained valuable polyphenols, such as glucoside isorhapontin (up to 12.0% of the dry extract), astringin (up to 4.6%), isorhapontigenin (up to 3.7%), and resveratrol glucoside piceid (up to 3.1%) (Figure 2) [68]. Considering that the bark extracts from this tree are rich in phenols, mostly in stilbenes, an extraction optimization strategy using a chemometric analysis to maximize the extraction of polyphenols was considered. The most suitable parameters allowing polyphenolic extraction were determined to be 80 °C and a ratio of bark/water of 50 mg/mL for obtaining low-molecular-weight polyphenols [68].

In addition to their antioxidant and antipsoriatic activity, extracts from this tree have been studied with regard to their in vitro antidiabetic potential [69]. For this, three different organs (needle, bark, and cone) of *P. mariana* were collected at different geographical locations, and their 80% ethanolic extracts were prepared [69]. All extracts protected PC-12-AC cells from glucose-induced toxicity. The total phenol content was higher in the cone extracts and lower in the needle extracts, which correlated with the antioxidant capacity, whereas this activity was not connected with the protective effect on glucose-induced toxicity [69]. The chemical characterization confirmed the presence in the extracts of terpenes and phenols, such as abietic acid, dehydroabietic acid, oxodehydroabietic acid, kaempferol, resveratrol, and taxifolin [69].

## 3. Challenges and Opportunities for Forest Extract Valorization

Forestry sectors worldwide have focused on the development and production of wood-based products on an industrial scale due to the importance of the timber industry and the existence of already established markets. However, in recent years there has been a trend toward promoting markets based on non-wood forest products within which the use of antioxidant and bioactive extracts is included [70].

Extracts from forest exploitation by-products, such as bark, needles, buds, leaves, knots, etc., given their richness in antioxidant and bioactive compounds, particularly phenols and terpenes, have a potential for developing new value-added cosmeceutical, nutraceutical, or pharmaceutical products. This is the case from promising extracts of forest species, such as *M. arboreus*, *A. rubrum*, and *P. mariana* analyzed above.

However, there are several potential problems in the application of these extracts in terms of (a) the management of by-products in the context of current forestry practices; (b) the low solubility, storage instability, and lack of selectivity towards a particular ROS of forest antioxidants; and (c) the lack of integration between the forestry sector and the cosmetic, alimentary, chemical, and food industries. Consequently, one of the big questions remaining is how to integrate the use of forestry extracts from by-products in a holistic process considering the current forestry practices.

Previous research using the bark of *P. mariana* attempted to answer this question. The bark is a primary by-product of wood processing often discarded as a forestry residue, burned for energy production, or even transformed into advanced carbon material. Intending to design innovative products using this residue, an integrative process using the *P. mariana* bark through the simultaneous incorporation of two different types of extraction (hydrodistillation and hot water extraction) was designed [71]. This method produced three natural extracts: the essential oil and hydrosol capturing the BS fragrance and the hot water extract enriched with antioxidant polyphenols (*trans*-isorhapontin, *trans*-resveratrol, *trans*-piceide, and *trans*-astringin). The remaining bark residue after retrieval of valuable chemicals still preserved its calorific value, which was almost identical to that of the non-extracted bark (approximately 20 MJ/Kg), indicating that BS bark would still be available for combustion, which is a common use in the context of existing practices [71]. This green process uses water as the only solvent, which can be recycled by cohobation (re-injection of condensate water into the still during hydrodistillation). An interesting aspect is that it can be extrapolated to the management of barks from other forest species, offering eco-responsible solutions to add value to these forest wastes through a biorefinery approach compatible with existing forestry procedures (Figure 5).

In recent years, it has been suggested that the inefficiency of natural antioxidants could be due to their low solubility, permeability, storage instability, first-pass metabolism, or gastrointestinal degradation [72]. Novel antioxidant delivery systems using forest extracts have been developed to counteract the low solubility, storage instability, and astringency of these extracts. The encapsulation of *Quercus resinosa* leaves infusion of submicron to nanometer size by spray-drying retained the good antioxidant capacity of the original infusion [73]. A multifunctional yogurt enriched with nanocapsules of *Quercus crassifolia* bark extract together with omega 6 and 3 [72] was also developed to mask the astringent taste of phenolics while conserving their antioxidant activity [74]. Encapsulation is based on the embedding effect of a polymer matrix, generating an environment in the capsule that protects the natural extract since it is capable of controlling the interactions between the interior and exterior sides. This is presented as a reasonable alternative to reduce the degradation and instability of antioxidant and bioactive molecules, such as polyphenols present in forest extracts, allowing their incorporation into functional foods and pharmaceutical products [72].

Another aspect that remains to be addressed in upcoming investigations is to define whether polyphenols and terpenes isolated from forest by-products or their metabolites are actually responsible for the antioxidant, antidiabetic, anti-inflammatory, or antipsoriatic activity. Although antioxidant methods using artificially generated radicals, such as DPPH, have served to test the activity of many forest extracts, their lack of biological relevance impedes making conclusions regarding their behavior as antioxidants in more complex biological systems. In the same way, the role of the metabolites of forest extracts on the signaling pathways involved in the diseases described above and their direct connection with oxidative stress should be more clearly defined in the future.

The use of bioactive molecules from forest resources for developing dietary supplements and nutraceuticals also faces several challenges. One of the challenges that have been pointed out to this type of product is the poor international regulations, which have led to unequal product quality requirements, thereby affecting their quality and safety with a negative impact on consumer confidence [75]. Many forest species are not regularly consumed as food, so the use of extracts and active fractions from forest by-products for developing new nutraceuticals or dietary supplements must include rigorous toxicological evaluations to identify risks associated with their regular use [76]. The research using forest extracts for nutraceuticals and dietary supplements is still in its early stages.

The French maritime pine bark extract is an example that illustrates how forest extracts from by-products can be successfully used to develop dietary supplements and cosmetic products. Maritime pine (*Pinus pinaster* Ait.) is native to the western Mediterranean, and its wood is used in the construction, furniture, and paper industries [77]. The procedure for obtaining the standardized water extract from pine bark was patented and is commercialized in various preparations under the trademark of Pycnogenol^®^, Oligopin^®^, and Flavagenol^®^ [78]. Pycnogenol^®^ was ranked among the 100 top-selling herbal dietary supplements in the United States in mainstream retail outlets [79]. Between 65% and 75% of its chemical composition is represented by polymeric procyanidins comprising catechin and epicatechin subunits with variable chain lengths, with other polyphenolic constituents, phenolic or cinnamic acids, and their glycosides also being described [80]. The success of this extract is explained by its diverse biological properties, which have been extensively revised [81,82,83,84], and its low toxicity [82,85,86]. It has also been used in various cosmetic preparations for its ability to improve visible cutaneous signs of aging [87,88,89]. The management of *Pinus pinaster* is sustainable. The trees are cultivated in monoculture in the southwest of France. The bark used for extraction is obtained from trees cultivated for 30 years, whose wood is used mainly by the pulp and paper industry [79]. Being a sustainable process, the cut trees are replaced by seedlings in a process controlled by forest rules and regulations [79,90].

The sustainable use of forest resources can be a challenging task. One of the risks associated with the commercial use of extractives is related to the over-exploitation of forests by communities, avoiding sustainable harvesting practices, particularly in countries with deficient forest regulation [70]. An example of this is the Himalayan ecosystems considered an important center of biodiversity, where the over-exploitation of forest resources, including the rampant removal of medicinal species coupled with the rapid threat of illegal trade, has resulted in widespread species extinction [91]. For example, the pepper bark tree (*Warburgia salutaris*), native to Southern Africa, has been widely harvested for the medicinal properties of its bark. Illegal collection, together with habitat degradation, has contributed to the fragmentation of populations and a severe decline in their distribution [92]. Before embarking on the development of a new product for nutraceutical, cosmeceutical, or pharmaceutical applications, the potential for sustainable exploitation of the promising forest species must be analyzed. For this, the characteristics of the habitat, the size of the population, the distribution patterns, as well as the reproductive mechanisms must be exhaustively examined [70].

Natural molecules have played a central role in the discovery of new drugs [93]. Although progress has been made in the investigation of bioactive molecules from forest resources allowing Taxol^®^ to be currently a successful approved drug, much remains to be done to promote an effective connection between the forestry and pharmaceutical industry. This requires understanding the challenges related to new drug discovery. Foremost, the pharmaceutical industry is subject to a wide range of government regulations and public policies. Furthermore, discovering new molecules has become much more rigorous and systematic in recent years, as “rational drug design” involves a conceptual understanding of the specific biological targets that must be “hit” in order to successfully treat a given disease. Bioactive molecules from forestry by-products could consequently have many obstacles to drug discovery. One of the most important challenges is the proper management of forest resources, allowing for the identification of interesting bioactive molecules, which in some cases are thermolabile, photosensitive, or can be degraded by hydrolysis. Although access to forest by-products could be extensive, if the molecule is found in very small quantities within the forest resource, having enough biological material to isolate and characterize can be difficult. Another obstacle could be obtaining intellectual property rights for isolated molecules with relevant pharmacological actions in their original form since natural compounds without any chemical modification cannot always be patented [93].

The need for more research in the area of chemical, biomedical, and forestry sciences is imperative and will require a synergistic collaborative framework between the wood industry with academia, the chemical, food, cosmetic and pharmaceutical industries. Research on business practices, strategic management, and marketing considering sustainability approaches, may help to address challenges, thereby stimulating the integration among the involved actors. Forests remain the most abundant sources of bioactive molecules on Earth, with innovative products based on them yet to be discovered and developed.

## Figures and Tables

**Figure 1 molecules-28-02045-f001:**
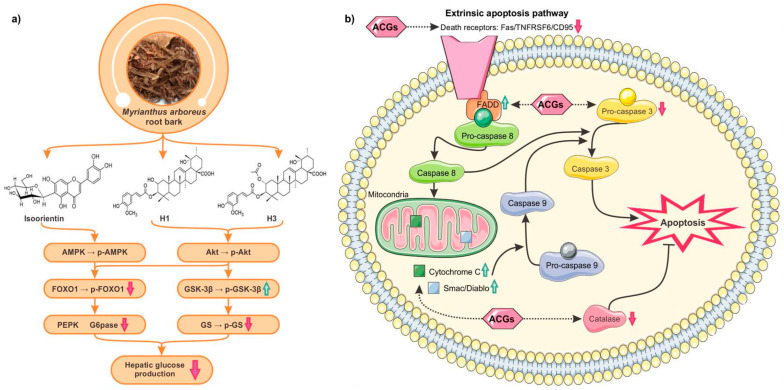
Mechanism of action of bioactive compounds from *Myriantus arboreus* root bark (**a**) and *Acer rubrum* bud (**b**) extracts according to in vitro experiments. (**a**) Compounds isolated from M. arboreous root bark, namely isoorientin, 3β-*O*-*trans*-feruloyl-2α,19α-dihydroxyurs-12-en-28-oic acid (H1) and 2α-acetoxy-3β-*O*-*trans*-feruloyl-19α-hydroxyurs-12-en-28-oic acid (H3), were determined to decrease the activity of hepatocyte glucose-6-phosphatase (G6pase) by increasing AMPK phosphorylation and to stimulate glycogen synthase (GS). This action led to the reduction of hepatic glucose production and the enhancement of glucose storage [44,45]. (**b**) Delayed death of neutrophils in tissues can cause an unwanted and exaggerated inflammatory reaction. Anhydroglucitol-core gallotannins (ACGs) from A. rubrum buds modulate the viability of human blood neutrophils. The significant increase of FADD, an adaptor protein that bridges proteins containing death receptors associated with a significant reduction of Fas/TNFRSF6/CD95 expression, suggests the implication of the extrinsic pathways in the induction of neutrophil apoptosis by ACGs. Neutrophils treated with ACGs showed a significant rise of SMAC/Diablo and cytochrome C, two caspase-dependent proteins released from mitochondria, thereby affecting the apoptotic process of human neutrophils [51].

**Figure 2 molecules-28-02045-f002:**
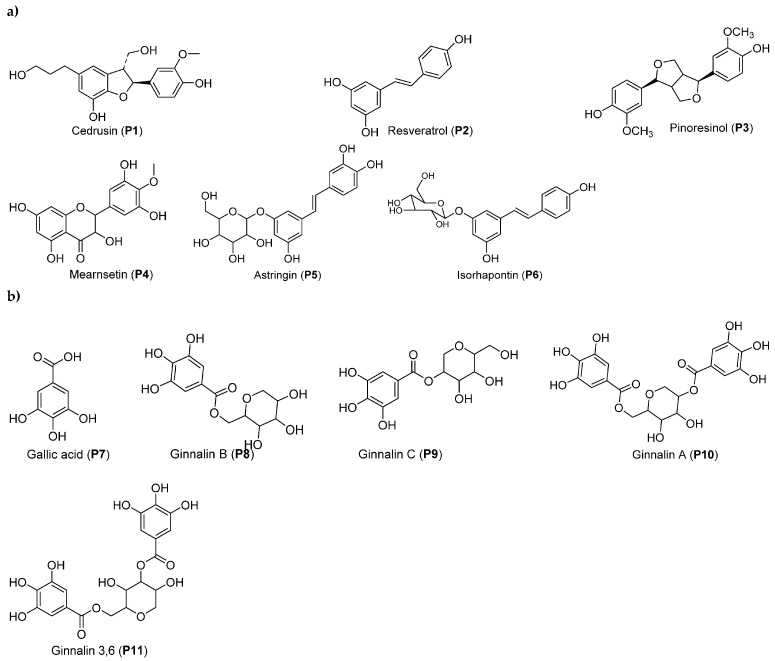
Main molecules identified in promising extracts from bark of black spruce (*Picea mariana*) (**a**) and red maple (*Acer rubrum*) (**b**).

**Figure 3 molecules-28-02045-f003:**
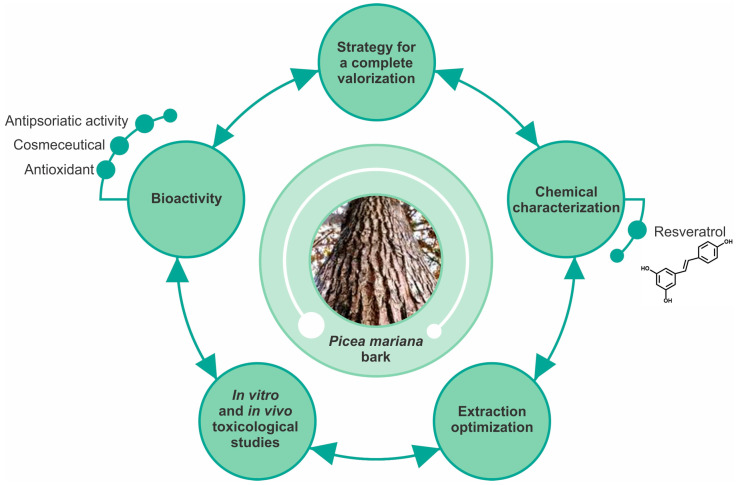
Strategy to valorize the *Picea mariana* bark extracts. The valorization of the bark extract from *P. mariana* involves several stages, starting from the chemical characterization and the identification of bioactive polyphenols, notably trans-resveratrol. The optimization of the extraction is crucial for guaranteeing reproducible yields of bioactive compounds. The most suitable parameters allowing polyphenolic extraction were determined to be 80 °C and a ratio of bark/water of 50 mg/mL especially for obtaining low-molecular-weight polyphenols. The toxicological studies are important for determining the safe concentrations for extracts both in vitro and in vivo. Extracts demonstrated to have potential activity as antiaging, antioxidant, and antipsoriatic agents.

**Figure 4 molecules-28-02045-f004:**
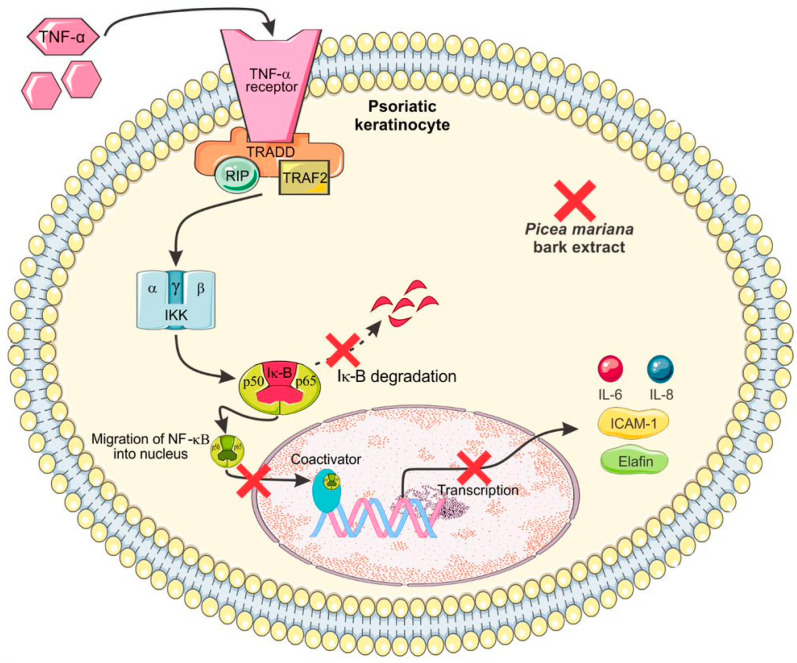
Mechanism of action of *Picea mariana* bark extract as a potential antipsoriatic treatment.

**Figure 5 molecules-28-02045-f005:**
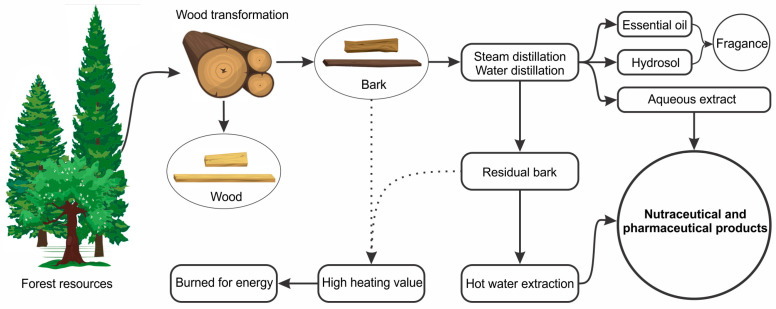
Proposed integrative process to obtain valuable extracts from *Picea mariana* bark in the context of existing forestry practices.

**Table 1 molecules-28-02045-t001:** Bioactive molecules identified in extracts from forest by-products of *M. arboreous*.

Molecule Name	Classification	Extract Type	Plant Tissue	Bioactivity	Ref.
Epicatechin	Flavonoid	Ethyl acetate fraction (EtOAc) of 70% ethanolic extract.	Stem bark	All compounds, except euscaphic acid, inhibited the in vitro action of α-amylase. Euscaphic acid stimulated the glucose uptake in C2C12 cells. Epigallocatechin, dulcisflavan, tormentic acid, and arjunolic acid showed hypoglycaemic and anti-hyperlipidaemic activities in treptozotocin (STZ)-induced diabetic rats. Dulcisflavan was considered the most active compound and an appropriate substrate for further drug development.	[38]
Epigallocatechin	Flavonoid				
Dulcisflavan	Flavonoid				
Euscaphic acid	ursane-type triterpenoids				
Tormentic acid	ursane-type triterpenoids				
Arjunolic acid	oleanane-type pentacyclic triterpenoid				
3β-*O*-*trans*-feruloyl-2α,19α-dihydroxyurs-12-en-28-oic acid (H1)	ursene-type pentacyclic triterpene	Ethyl acetate fraction (EtOAc) of 95% ethanolic extract.	Root bark	All compounds decreased in vitro the activity of hepatocellular glucose-6-phosphatase (G6Pase) and activated glycogen synthase via the phosphorylation of glycogen synthase kinase-3. The compound (H3) and isoorientin were determined to be the most potent in modulating glucose homeostasis in liver cells.	[39,40]
2α-acetoxy-3β-*O*-*trans*-feruloyl-19α-hydroxyurs-12-en-28-oic acid (H3)	ursene-type pentacyclic triterpene				
ursolic acid	pentacyclic triterpene				
isoorientin	C-glycosylflavone				
orientin	C-glycosylflavone				
3,4-dihydroxybenzaldehyde	phenolic aldehyde				
3β,6β-dihydroxyolean-12-en-29-oic acid (myrianthinic acid)	pentacyclic triterpene	Ethyl acetate fraction (EtOAc) of methanolic extract.	Stem bark	-	[41,42]
2β/3β, 24-trihydroxy-olean-12-en-28-oic acid (arboreic acid)	oleanane-type triterpenoid				
protocatechuic acid	phenolic acid	70% methanolic extract (MAL)	Leaves	MAL administration significantly reduced body weight gain, basal glycemia, and insulin resistance in mice receiving a high-fat diet (HFD). MAL significantly downregulated the mRNA expression of IL-6, IL-1β, and TNF-α, known as obesity-associated inflammatory markers. MAL improved the altered expression of adipokines (leptin and adiponectin) in obese mice.	[43]
methylumbelliferone fucopyranoside	hydroxycoumarin				
tectoridin	glycosyloxyisoflavone				
vanillic acid	phenolic acid				
medicagenic acid	triterpenoid				
brahmic acid	pentacyclic triterpenoid				
arjunolic acid	oleanane-type pentacyclic triterpenoid				

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
