# Peer review of "Bioactive Molecules from *Myrianthus arboreus*, *Acer rubrum*, and *Picea mariana* Forest Resources"

_molecules, 2023, doi:10.3390/molecules28052045_

Round 1
Reviewer 1 Report
Comments to authors are as follows:
1. It is suggested to the authors to change the title as the manuscript primarily highlights the promising forest resources of Myrianthus arboreus, Acer rubrum and Picea marina.
2. Authors need to highlight the reason why three species were stated or valued as promising forest resources.
3. There some glaring errors on scientific name and acronyms.
4. It is suggested that the heading of subtopic 3 to be re-arranged according to the flow of content in the manuscript.
Author Response
Dear Dr. Chen
Thanks for allowing us to submit a revised draft of the manuscript “Bioactive molecules from Myrianthus arboreus, Acer rubrum, and Picea mariana forest resources” (molecules-2134973). We appreciate the time and effort of the reviewers and are grateful for the constructive comments and valuable improvement of our paper.
We have incorporated most of the suggestions made by the reviewer. Please, find below the point-by-point responses to these comments. We hope that this revised version will be of your satisfaction.
We look forward to the outcome of your assessment.
Yours sincerely,
PS: Please see the revised manuscript in the attachment.
Reviewer 1
Comments to authors are as follows:
- It is suggested to the authors to change the title as the manuscript primarily highlights the promising forest resources of Myrianthus arboreus, Acer rubrum and Picea marina.
R) The title of the manuscript was changed as suggested
- Authors need to highlight the reason why three species were stated or valued as promising forest resources.
R) Following this suggestion, in the last paragraph of the Introduction section, the reasons for choosing these forest species are highlighted.
- There some glaring errors on scientific name and acronyms.
R) In the revised manuscript, the errors on scientific names and acronyms have been corrected.
- It is suggested that the heading of subtopic 3 to be re-arranged according to the flow of content in the manuscript.
R) Following this suggestion, the heading of subtopic 3 was modified.
Reviewer 2 Report
[Molecules] Manuscript ID: molecules-2134973 : Bioactive Molecules from Forest Resources
title : Bioactive Molecules from Myrianthus arboreus, Acer rubrum, and Picea mariana Forest Resources
is more suitable for this submission as the authors focus on these resources anyway.
L12 ...widely recognized for their antioxidant activity. >>>widely recognized for their biological activities.
L13 ..are found at high concentrations in forest by-products .. >>> this sentence is obsolte.. delete it! as amounts may vary!
L14.. This review provides an updated analysis of antioxidant... >> The present literature review focuses on the bioactives from the phytochemicals of
Myrianthus arboreus, Acer rubrum, and Picea mariana forest resources and by-products covering the last 2 decades.
L16 .. and medicinal herbaceous plants used in folk medicine. >>> should be removed from the whole work to enable the reader to concentrate on forest resouces
so please reformulate the whole mans in this aspect.
L18-9 ...prevention and/or treatment of diseases related to oxidative stress such as diabetes, inflammation, and psoriasis...> do not excist on evidence basis from natural product drugs yet!
so remove all claims on prevention - treatment as it is pharmaceutical related, which are approved from the Ministry of Health. Other in vitro
in silico data are just misleading information. Please consult a Pharmacist on the drug related informations. Moreover, "diabetes, inflammation, and psoriasis"
medical therapies in the modern medicine is limited, and there is NO FDA and EMA approved natural drug or therapy to best of my knowledge with evidence based data.
L20 .. forest residues .. >>> just use one terminology: forest by-products all over the mans. othervise, artefacts, residues, junk.. etc will again mislead..
L21... functional foods... >>> the authors should focus on this topic, rather then "active pharmaceutical ingredients" such as "taxol" or have a separate chapter on such products
supported with the expertise of a pharmacist and pharmacognosist. Also cosmetics, spices and culinary aspects would have a good focus and much data.
L25 : drug > it is suggested to remove the aspect of pharmaceuticals and drugs from this review without the contributions of a pharmacist / pharmacognosist
in the introduction
reference 1 has relatively high impact, is this the only primary source for this 2-3 paraghraphs? check it out carefully.
reference 2: Luca et al. Bioactivity of Dietary polyphenols: The Role of Metabolites. Critical Reviews in Food Science and Nutrition 2020
is not the primary reference for diseases such as "cancer, inflammation, brain neuromodulation, Alzheimer’s disease, diabetes, and psorias"
as it is refered to.. also not related to therapeucticals!
L60-2 statements are also is scientifical not correct! so either remove or revise without speculative claims.
L62-8 statement refering 4 as a source is also not the primary source! Prefer to refer a forestry secondary metabolites biosynthesis book, chapter or review(s) instead
L78:This review provides an updated analysis of the presence of bioactive>>> literature review should focus on the utilization except medicinal ones .. analysis also misleading?
explain which analyses? which methods???
L79 ...terpenoids, covering also oxygenated derivatives> terpenoids= oxygenated derivatives??? please use correct terminology! check!
L81: ...and medicinal herbaceous plants used in folk medicine .. >> this will be too broad and is a topic for review itself! Forest MAPS...
I strongly exclude this topic along with medicines!!!
L85: ...or prevention and/or treatment of diseases related..>>>> as explained above, there is no point to have a mixture of different topics.
L87: .. nutraceutical ..> the authors should concentrate on feed, food additives, flavors etc.. but NOT in pharmaceuticals, and therapeuticals as this is another field of expertise and review!
L89: herbaceous medicinal plants, and healthy..> MUST be excluded in the context of this review to be accepted for publication.
...
L153: Table 1. Content of bioactive polyphenols in healthy foods, medicinal plants, and forest trees.>>> should be Table 1. Content of bioactive polyphenols in forest trees
overall, the work submitted needs thorough revisions before acceptance.
Author Response
Dear Dr. Chen
Thanks for the invitation to submit a revised draft of the manuscript “Bioactive molecules from Myrianthus arboreus, Acer rubrum, and Picea mariana forest resources” (molecules-2134973). We appreciate the time and effort of the reviewers and are grateful for the constructive comments contributing to a valuable improvement of our paper.
We have incorporated most of the suggestions made by the reviewer. Please, find below the point-by-point responses to these comments. We hope that this revised version will be of your satisfaction.
Following the substantial changes that we have made in answer to the reviewers comments, we hope that our manuscript is now suitable for publication
Yours sincerely,
PS: In attachment, please see the revised manuscript.
Reviewer 2
Title : Bioactive Molecules from Myrianthus arboreus, Acer rubrum, and Picea mariana Forest Resources is more suitable for this submission as the authors focus on these resources anyway.
R) The title of the manuscript was changed as suggested
L12 ...widely recognized for their antioxidant activity. >>>widely recognized for their biological activities.
R) Thanks. The proposed change was made in the Abstract
L13 ..are found at high concentrations in forest by-products .. >>> this sentence is obsolte.. delete it! as amounts may vary!
R) This phrase now reads as follows: “These molecules are found in forest by-products such as bark, buds, leaves, and knots, commonly ignored in forestry decisions.”
L14.. This review provides an updated analysis of antioxidant... >> The present literature review focuses on the bioactivities from the phytochemicals of Myrianthus arboreus, Acer rubrum, and Picea mariana forest resources and by-products covering the last 2 decades.
R) The proposed change was introduced in the Abstract
L16 .. and medicinal herbaceous plants used in folk medicine. >>> should be removed from the whole work to enable the reader to concentrate on forest resources, so please reformulate the whole mans in this aspect.
R) This phrase was removed from the Abstract as suggested
L18-9 ...prevention and/or treatment of diseases related to oxidative stress such as diabetes, inflammation, and psoriasis...> do not excist on evidence basis from natural product drugs yet! so remove all claims on prevention - treatment as it is pharmaceutical related, which are approved from the Ministry of Health. Other in vitro in silico data are just misleading information. Please consult a Pharmacist on the drug related informations. Moreover, "diabetes, inflammation, and psoriasis" medical therapies in the modern medicine is limited, and there is NO FDA and EMA approved natural drug or therapy to best of my knowledge with evidence based data.
R) We agree with the reviewer that there is no FDA or EMA-approved natural product for the treatment or prevention of these conditions. Consequently, in the Abstract (L18-9) we eliminated the phrase that established that these natural extracts are used for the prevention and/or treatment of these diseases.
L20 .. forest residues .. >>> just use one terminology: forest by-products all over the mans. othervise, artefacts, residues, junk.. etc will again mislead..
R) As suggested, the term forest by-product was used throughout the manuscript
L21... functional foods... >>> the authors should focus on this topic, rather then "active pharmaceutical ingredients" such as "taxol" or have a separate chapter on such products supported with the expertise of a pharmacist and pharmacognosist. Also cosmetics, spices and culinary aspects would have a good focus and much data.
R) The idea of this review is to show the diversity of bioactive molecules (terpenes and phenols) occurring in forest resources, mainly in by-products of Myrianthus arboreus, Acer rubrum, and Picea mariana which could potentially serve for the future development of functional foods and therapeutic products and the challenges that further development would entail. We agree with the reviewer that there is still a critical need to demonstrate the pharmacological effects of such extracts in animal models and humans, but we believe that within forest by-products there is great potential for the development of the pharmaceuticals that deserve to be known and investigated. For instance, Taxol® remains a successful case for a natural product from forests used in the contemporary pharmacopeia. Showing this potential is important for the development and innovation of the industrial forestry sector, which produces millions of tons of waste per year, which are burned as supplementary energy sources. Additionally, for the pharmaceutical industry, a higher interest in forest by-products could represent an opportunity to discover new bioactive molecules.
L25 : drug > it is suggested to remove the aspect of pharmaceuticals and drugs from this review without the contributions of a pharmacist / pharmacognosist in the introduction.
R) Considering that this review does not focus on drug development, but on showing the therapeutic potential of forest extracts, we removed the word drug from the keywords as suggested by the reviewer.
reference 1 has relatively high impact, is this the only primary source for this 2-3 paraghraphs? check it out carefully.
R) Following this suggestion, we included references 2, 3, and 4.
reference 2: Luca et al. Bioactivity of Dietary polyphenols: The Role of Metabolites. Critical Reviews in Food Science and Nutrition 2020 is not the primary reference for diseases such as "cancer, inflammation, brain neuromodulation, Alzheimer’s disease, diabetes, and psorias" as it is refered to.. also not related to therapeucticals!
R) We thank the reviewer for this observation. Given the amount of original preclinical and clinical research on the biological activity of polyphenols in diseases such as cancer, inflammation, brain neuromodulation, Alzheimer's disease, diabetes, and psoriasis, we believe that the article by Luca et al. makes an excellent analysis of the importance of polyphenol metabolites in the intrinsic biological effects reported for parent polyphenols recognized for their bioactivity (curcumin, resveratrol, quercetin, rutin, etc.), which could help readers understand the low bioavailability/high bioactivity paradox described for these molecules. However, we agree with the reviewer that this single reference does not show the impact of these molecules on such diseases. Therefore, we decided to include another reference (Rahman et al. Molecules 2022, 27, 233) that shows in a summary form the importance that polyphenols could have in these diseases.
L60-2 statements are also is scientifical not correct! so either remove or revise without speculative claims.
R) As suggested, the sentence has been re-written as follows: “In forest trees, these compounds are found in a higher proportion in forest by-products such as barks and knots than in wood”
L62-8 statement refering 4 as a source is also not the primary source! Prefer to refer a forestry secondary metabolites biosynthesis book, chapter or review(s) instead
R) As suggested, the reference was changed to a book chapter as follows: “Umezawa, T. Chemistry of Extractives. In Wood and Cellulosic Chemistry, Second Edition, Revised, and Expanded; CRC Press, 2000 ISBN 978-0-8247-0024-9.”
L78:This review provides an updated analysis of the presence of bioactive>>>literature review should focus on the utilization except medicinal ones .. analysis also misleading? explain which analyses? which methods???
R) Following this observation, the purpose of this literature review was changed (please see the last paragraph of the Introduction section)
L79 ...terpenoids, covering also oxygenated derivatives> terpenoids= oxygenated derivatives??? please use correct terminology! check!
R) In order to avoid confusion between the terms terpenes and terpenoids, we have preferred to use terpenoids in the revised manuscript. However, the two terms have been defined as follows: ‘’Terpenes represent a heterogenous class of natural products with more than 70,000 structures reported so far [49]. The structure of terpenes is based on the linkage of isoprene units classified as monoterpenes (2 isoprene units), sesquiterpenes (3 units), diterpenes (4 units), triterpenes (6 units), tetraterpenes (8 units). In contrast to terpenes that are simple hydrocarbons, terpenes containing additional functional groups, usually oxygen-containing are called terpenoids [50]. Thus, terpenoids can be differentiated from one another by their basic skeleton and functional groups.’’
L81: ...and medicinal herbaceous plants used in folk medicine .. >> this will be too broad and is a topic for review itself! Forest MAPS... I strongly exclude this topic along with medicines!!!
R) As explained above, in the interest of improving the understanding of the readers, the wording of this paragraph including the purposes of the review was modified (please see the last paragraph of the introduction).
L85: ...or prevention and/or treatment of diseases related..>>>> as explained above, there is no point to have a mixture of different topics.
R) As stated above, we agree with the reviewer that there is no FDA or EMA-approved natural product for the treatment or prevention of these conditions, so this phrase was deleted.
L87: .. nutraceutical ..> the authors should concentrate on feed, food additives, flavors etc.. but NOT in pharmaceuticals, and therapeuticals as this is another field of expertise and review!
R) This review has tried to show the diversity of molecules that can be found in forest resources, particularly in by-products derived from wood exploitation (barks, stems, leaves, shoots, etc.). These molecules are also found in healthy foods and herbaceous medicinal plants, therefore, we believe there is an underlying potential for their therapeutic use. Although we agree with the reviewer that it is not currently possible to recommend the use of these extracts for disease prevention and treatment, as both preclinical and clinical investigations must be completed, we believe that there is a therapeutic potential in these forest extracts that deserves to be recognized and further investigated. In fact, the boreal forest of North America and forests in Africa are home to several hundred thousand aboriginal people who have been using the forest species here revised (P. mariana, M.arboreus, and A.rubrum) in traditional healthcare systems for thousands of years (Bi et al., 2016; Bobuya et al., 2022; Uprety et al., 2012; Zhang et al., 2015). Consequently, the therapeutical potential use of such extracts not only has been recognized by traditional cultures but also by other investigations here reviewed (Dickson et al., 2016; Diouf et al., 2009; García-Pérez et al., 2012; Kasangana et al., 2019; Zhang et al., 2015). As we recognize that many challenges currently exist before these extracts can be used as functional foods or therapeutic products, these issues are discussed in section 4.
L89: herbaceous medicinal plants, and healthy..> MUST be excluded in the context of this review to be accepted for publication.
R) As suggested, the name of section 2 changed (L89).
L153: Table 1. Content of bioactive polyphenols in healthy foods, medicinal plants, and forest trees.>>> should be Table 1. Content of bioactive polyphenols in forest trees.
R) The name of Table 1 was changed as suggested by the reviewer.
Overall, the work submitted needs thorough revisions before acceptance.
R) We thank the reviewer for these suggestions which significantly improved our manuscript.
Round 2
Reviewer 2 Report
Thank you for the improvements
however, some minor revisions are still needed.
to be a drug canditate, the chemistry has to comply to pharmacopoeia.
any chemical which is even not close to this point of view, cannot be a candidate. Thus the medicinal claims should be considered in biological activity level esp in vitro and in silico data!!!
As you know there is NO drug for diabetes; only Metformin a biguanide antihyperglycemic agent and insulin treatment is applied..
Thus to state that forest products are antidiabetic is NOT scientifically correct.. there may be traditional application, but no modern medical application is to be claimed.
L15 ..focuses on the bioactivities (antioxidant,...> focuses on in vitro experimental bioactivity such as antioxidant,...... with the potential for further drug development.
L91 - 5: Why there is a need to define the chemistry of polyphenols?? Avoid trivial book information! also elsewhere..
L158 - 62: Why there is a need to define the chemistry of TERPENES?? abc?
L148 Table 1. Content of bioactive polyphenols in forest trees.
I do not agree that Ranunculus macrophyllus has 131.2mg hesperidin! this is not a scientific view! according to biosythetic factors 0-200 mg or more?? depending... which is also the case for other secondary metabolites.. the table is obsolete.. the data provided is from single random selected publications which do not necessary reflect the real data. and there is many questions of the reported work, most of the data is not validated!!!
The intention is good, but data of the table MUST be either simplified or should have ranges..
What does "Healthy Food" mean??? further more Ranunculus macrophyllus is not an edible MAP? so the table has many inconsistencies.
My personal suggestion is JUST to concentrate on Wood /Forest Products rather on other MAPs.. Also if content is desparate needed, this must be a RANGE, such in nature, according to validated qunatifications!!!
same applies for
Table 2. Content of some bioactive terpenoids in forest trees
L291...The antidiabetic potential >... it is important to write that this or similar work is IN VITRO/ in vivo / in silico!
L 390 ..Figure 1. Mechanism of action..> give detials on the reference it was used.. 96??? also in vitro data!!!!
L292 ..Inflammation is a primary mechanism available ..>>> why is this specific data needed within this review??? This is also specific data very limited related to "Bioactive Molecules from Myrianthus...." only brief data on mechanisms are more than enough.. The chemistry and activity associated in a short way would be more efficient in my opinion?
Not only to inflammation, to other biological activity as well in the "review", this point of view is strongly suggested!
fig 2 compounds should have an identifier by number continuing from fig 1 on.. the whole mans should have consistent numbering of molecules illustrated throughout.
L 470 >barks??? > bark!!!
L472.. Diet-Fed Mice should be mentioned..
L473..appears to be useful to counteract obesity and gut dysbiosis...> this sentence is also not scientific
L484..Psoriasis is a common.. > but not the core point of this review!!! one paragraph info on the pathology is not necessary.. as highlighted before, this review should focus on the chemistry and botany with evidence based valideated data..
L615.. why there is a need to integrate for phase 1 preclincal research "pharmacodynamic and pharmacokinetic characteristics" of forest extractives??? For clinical studies there is more to do.. and not the part of this phytochemistry review!
4. Challenges and opportunities for forest antioxidant extracts valorization
what about "pycnogenol" rather taxol? it is would be more in line with this review...
Finally, the authors should carefully review the work in the light of pharmacognosy / food additives rather than APIs. The work needs still revisions prior acceptance.
Author Response
Dr. Ada Chen
Assistant Editor
Molecules
Subject: Revision of the manuscript “Bioactive molecules from Myrianthus arboreus, Acer rubrum, and Picea mariana forest resources” (molecules-2134973)
Dear Dr. Chen
Thanks for allowing us to submit a revised draft of the manuscript “Bioactive molecules from Myrianthus arboreus, Acer rubrum, and Picea mariana forest resources” (molecules-2134973). We appreciate the time and effort of reviewer 2 and are grateful for the constructive comments and valuable improvement of our paper.
We have incorporated all of the suggestions made by the reviewer. Please, find below the point-by-point responses to these comments. We hope that this revised version will be of your satisfaction.
We look forward to the outcome of your assessment.
Yours sincerely,
Dr. Tatjana Stevanovic
Renewable Materials Research Center (CRMR),
Department of Wood Sciences and Forestry.
Université Laval, Québec city, QC, Canada.
tatjana.stevanovic@sbf.ulaval.ca
